# Core clock genes adjust growth cessation time to day-night switches in poplar

Daniel Alique ®[1], Arturo Redondo López[1], Nahuel González Schain[1,2], Isabel Allona ®[1,3], Krzysztof Wabnik ®[1,3] ✉ & Mariano Perales ®[1,3] ✉

Poplar trees use photoperiod as a precise seasonal indicator, synchronizing plant phenology with the environment. Daylength cue determines *FLOWERING LOCUS T 2* (*FT2*) daily expression, crucial for shoot apex development and establishment of the annual growing period. However, limited evidence exists for the molecular factors controlling *FT2* transcription and the conservation with the photoperiodic control of *Arabidopsis* flowering. We demonstrate that *FT2* expression mediates growth cessation response quantitatively, and we provide a minimal data-driven model linking core clock genes to *FT2* daily levels. *GIGANTEA* (*GI*) emerges as a critical inducer of the *FT2* activation window, time-bound by *TIMING OF CAB EXPRESSION* (*TOC1*) and *LATE ELONGATED HYPOCOTYL* (*LHY2*) repressions. CRISPR/Cas9 loss-of-function lines validate these roles, identifying TOC1 as a long-sought *FT2* repressor. Additionally, model simulations predict that *FT2* downregulation upon daylength shortening results from a progressive narrowing of this activation window, driven by the phase shift observed in the preceding clock genes. This circadian-mediated mechanism enables poplar to exploit FT2 levels as an accurate daylength-meter.

In boreal and temperate regions, trees undergo an annual alternation in the shoot apex between active growth and dormancy. Proper timing of growth cessation in autumn, which precedes dormancy, is crucial for their adaptation, thus synchronizing seasonal phenology with environmental permissiveness for growth. Trees like poplar rely on photoperiod as the most accurate cue to set the annual vegetative growing period, ceasing growth when the daylength falls below a threshold known as the critical daylength. Days with hours of light above the critical daylength are referred to as long-day (LD), while those below are called short-day (SD)[1,2]. Daylength input is perceived in leaves, and it sets the ~24 h oscillations of the circadian clock that ultimately converge to control the daily activation of photoperiodic effector FLOWERING LOCUS T 2 (FT2) (Fig. 1a)[3]. Under LD, FT2 daily expression is essential for promoting shoot apex vegetative

development, whereas the SD induces FT2 downregulation, marking the onset of growth cessation irrespective of the tree's age[4–6]. Therefore, control of the annual growing period in poplar depends on the photoperiodic pathway rather than age-dependent regulators.

In poplar, several photoperiodic-controlled genes have been proven to regulate *FT2* transcription. *LATE ELONGATED HYPOCOTYL 2 (LHY2)* binds to 3'UTR of *FT2* and is necessary for FT2 downregulation during nightlength extension. Accordingly, LHY2 overexpression reduces *FT2* level, while *LHY2* RNAi increases it, delaying poplar growth cessation[5,7]. A similar delay is observed in the RNAi of *TIMING OF CAB EXPRESSION (TOC1)*[7]. In *Arabidopsis* orthologs, *toc1-1* knockout increases *FT* mRNA expression, accompanied by a slight phase shift of *CONSTANS (CO)* transcription to the daytime[8]. Moreover, TOC1 is a general transcriptional repressor that targets TGTG motifs through its

[1]Centro de Biotecnología y Genómica de Plantas (CBGP, UPM-INIA) Universidad Politécnica de Madrid (UPM) - Instituto Nacional de Investigación y Tecnología Agraria y Alimentaria (INIA, CSIC), Campus de Montegancedo, Pozuelo de Alarcón, 28223 Madrid, Spain. [2]Instituto de Biología Molecular y Celular de Rosario, CONICET, Facultad de Ciencias Bioquímicas y Farmacéuticas, Universidad Nacional de Rosario, Rosario, Argentina. [3]Departamento de Biotecnología-Biología Vegetal, Escuela Técnica Superior de Ingeniería Agronómica, Alimentaria y de Biosistemas, Universidad Politécnica de Madrid (UPM), Madrid 28040, Spain. ✉e-mail: k.wabnik@upm.es; mariano.perales@upm.es

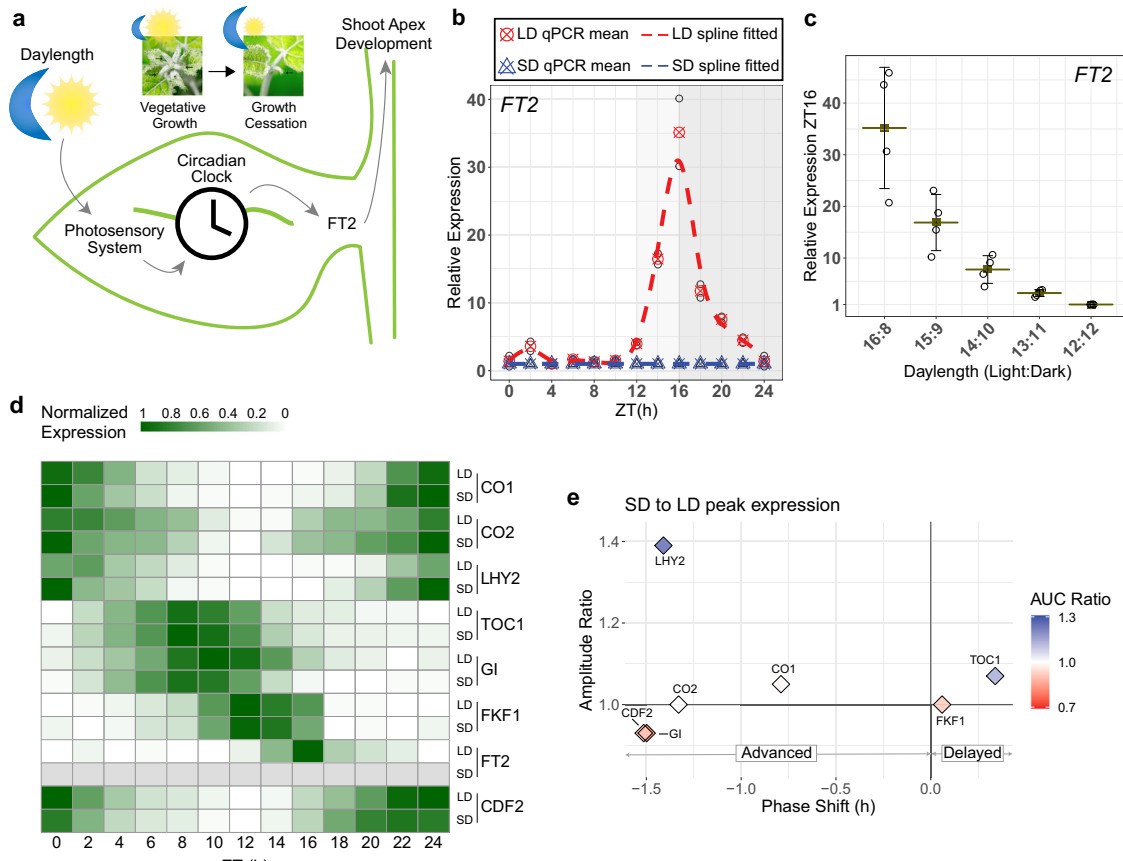

**Fig. 1 | Daylength modulates the expression of FT2 and its transcriptional regulators to control poplar shoot apex growth. a** Overview of leaf-localized daylength sensing mechanism that controls the transition from active growth to growth cessation in poplar. **b** Daily *FT2* transcription in LD (circles, red) and SD (triangles, blue). Spots represent qPCR results, with means of the two biological replicates used for fitting highlighted. Dashed lines depict a polynomial spline fit. *FT2* in SD was set to 1 due to nondetection. **c** *FT2* expression measured by qPCR at ZT16 under different daylengths. Mean ± sd, *n* = 4 biological replicates. **d** Heat map of daily transcription patterns for the photoperiodic-responsive genes in LD and SD. qPCR means (b and Supplementary Fig. 2) normalized 0–1 per gene. *FT2* in SD is shown in grey due to nondetection. **b**–**d** ZT, Zeitgeber Time in hours. **e** SD to LD deviation of the gene expression patterns shown in (**d**) for phase (time of maximal gene expression) shift, amplitude ratio, and daily total transcript ratio (AUC−Area Under the Curve) after Gaussian fit (Supplementary Fig. 2).

CCT DNA-binding domain[9]. This intermolecular binding is stabilized forming a trimeric complex with NUCLEAR FACTOR Y subunits B and C (NF-YB/C)[10]. Interestingly, this mode of transcriptional regulation by TOC1 is analogous to the one previously described for CO activation of *FT*. The CCT domain of CO mediates its binding to two CORE sites− TGTG(N2-N3)ATG, sharing the TGTG sequence, in the proximal *FT* promoter[11], and this is stabilized by CO/NF-Y interaction[12–14]. *Arabidopsis* TOC1 and CO were also reported to be associated in vivo[15].

Another circadian gene, *GIGANTEA* (*GI*), is critical for *FT2* expression. *GI* RNAi drastically decreases *FT2* in LD, triggering growth cessation. Conversely, GI overexpression upregulates *FT2*, and growth cessation under SD is delayed[4]. In *Arabidopsis*, GI promotes daily *FT* transcription through two different pathways: 1) CO-mediated activation; GI enhances the induction of *CO* transcription[16], and interacts with FLAVIN-BINDING, KELCH REPEAT, F-BOX 1 (FKF1), which ubiquitinates *CO* repressors CYCLING DOF FACTORS (CDFs) for degradation[17]. 2) CDF-mediated repression; the CDF family regulates *CO* and directly represses *FT*[18]. This GI-FKF1-CDF module seems to be conserved in poplar, where CDF overexpression leads to *FT2* down-regulation, advancing poplar growth cessation in SD[4]. Furthermore, GI is recruited to *FT* promoter regions in both *Arabidopsis* and poplar, pointing to direct *FT* regulation independent of the mentioned pathways[4,19].

Along with GI, CO is the key factor promoting *FT* in Arabidopsis. CO is a night-sensitive protein active at dusk in LD but degraded in SD

due to night advance[20]. In poplar, CO orthologs CO1/2 promote *FT2*, as *CO1/2* RNAi reduces *FT2* level and leads to early growth cessation in SD[21]. Intriguingly, a CO-independent repressor pathway dominates in poplar. The overexpression of either CO1 or CO2 does not upregulate *FT2* expression under SD, unlike their homolog in *Arabidopsis*[22–24].

Despite a substantial body of evidence on individual roles of photoperiodic-responsive genes in modulating *FT2* expression, significant gaps persist in understanding how the circadian clock collectively shapes *FT2* regulation in response to environmental cues such as daylength. *Arabidopsis* provides insights into the genetic pathways regulating *FT* in flowering, which bears significant similarities and could potentially aid in our knowledge. Nevertheless, the extent of conservation between the two systems and phenological processes remains uncertain. Moreover, current models of the gene networks controlling flowering in *Arabidopsis* yield predictions that do not agree with observed *FT* mRNA levels after manipulating core clock genes LHY2 and TOC1, implying there are missing factors in our comprehension of FT2 regulation (Supplementary Fig. 1)[25,26].

To address and refine the current model of FT control, we used CRISPR genetics, temporal transcriptional analysis, and computer simulations in the poplar model system *Populus tremula x P. alba*. By comparing different daylength conditions, we reveal distinct roles for core clock components in tailoring temporal FT2 expression patterns in response to environmental signals. A minimal core system is composed of two time-shifted repressors and a general activator.

Furthermore, we pinpoint mechanistic similarities and differences between *Populus* and *Arabidopsis*.

## Results

### FT2 and photoperiodic regulators of shoot apical growth expression patterns comparing LD with SD

To investigate how *FT2* expression level is regulated by daylength in poplar, we first recorded the daily expression pattern of *FT2* mRNA. In LD (16 h light: 8 h dark) condition, *FT2* showed a strong peak of expression at ZT16 (ZT refers to Zeitgeber Time, i.e. hours from dawn) in the intersection between day and night. Nevertheless, when exposed to SD condition (12 h light: 12 h dark), *FT2* transcript could not be detected (Fig. 1b). Furthermore, the maximal level of *FT2* expression was gradually decaying as the daylength shortens (Fig. 1c). These findings indicate that *FT2* level is tightly controlled by the duration of the day. To further study a molecular mechanism behind this *FT2* control by daylength, we studied the expression kinetics of core circadian photoperiodic regulators, by comparing their expression levels in SD and LD conditions after fitting their daily patterns to a Gaussian curve. We observed a significant advance >1.5 h of phase, defined as the time of maximal expression, for *GI* and *CDF2*, followed by *CO2* (1.3 h) and *CO1* (0.8 h) in SD condition (Fig. 1d, e, and Supplementary Fig. 2). Similarly, *LHY2* transcription advanced 1.4 h, with a nearly 1.4-fold increase in its amplitude. Conversely, *FKF1* showed minimal variations, like *TOC1*, which slightly increased the total accumulated transcript 1.1 times. Interestingly, TOC1 appeared later than *LHY2* in both conditions, preceding the *FT2* expression peak in LD (Fig. 1d, e and Supplementary Fig. 2).

Notably, these observations contrast with results reported for *Arabidopsis* homologs, in which these core circadian clock genes, including *TOC1*, exhibit a widespread phase advance when transitioning to SD (Supplementary Fig. 3)[27]. Also, unlike in *Arabidopsis*, poplar does not show an increase in *GI* level in the SD condition (Fig. 1e and Supplementary Fig. 3b). Furthermore, when contrasting *GI* and *TOC1* relative phases under LD, *GI* is expressed earlier than *TOC1* in *Arabidopsis*, whereas this order is reversed in poplar. The phase advances described in SD keep *GI* preceding *TOC1* expression in *Arabidopsis*, while in poplar, both expression patterns coincide with a phase ~ ZT8.5 (Fig. 1d, e and Supplementary Fig. 3). Finally, poplar *CO1/2* exhibit a significantly delayed expression phase compared to *Arabidopsis* CO, both in LD (~5 h later) and SD (~8 h later), displaying their maximum expression at the beginning of the day (Fig. 1d, Supplementary Figs. 2a, b, and 3a).

### CRISPR-Cas9 lines unveil the role of TOC1, GI, and LHY2 in regulating FT2 transcription

To further dissect the role of the core circadian clock genes in poplar growth cessation and *FT2* regulation, we characterized the CRISPR-Cas9 loss-of-function lines for LHY2[5], TOC1, and GI (see Supplementary Fig. 4). In LD condition, WT plants produced 3 young leaves at the apex (Fig. 2a). In contrast, *gi* mutant fails to reach this stage of full growth and ceases growth one month after being transferred to soil in LD, a response reminiscent of *ft2* knockout (Fig. 2a). Conversely, *lhy2* and *toc1* trees, when compared to WT, require a shorter daylength (14 h instead of 15 h) to start reducing the activity of shoot apical growth and instead remain actively growing under SD condition (Fig. 2a).

This growth cessation phenology in *Populus* tightly correlates with *FT2* expression level observed in LD condition. In *gi*, *FT2* mRNA remains undetected, whereas in *lhy2* and *toc1*, there is a similar significant increase in both amplitude (up to 2.4-fold and 2.0-fold, respectively) and total daily transcript accumulation (up to 2.4-fold and 2.3-fold, respectively) compared to WT (Fig. 2b–d). Moreover, the expression phase of *FT2* is markedly advanced by 1.1 h in *lhy2* and 1.5 h in *toc1* mutants (Fig. 2b–d). Regarding the daily patterns of *FT2* photoperiodic regulators, in *lhy2*, it is noteworthy the phase advance for *GI* (4.2 h), *CO1*

(9.2 h), *CO2* (8.6 h), and *CDF2* (7.0 h) (Fig. 2c, d, and Supplementary Fig. 5). However, *TOC1* shows a minor delay and reduction in amplitude (0.6-fold). In *toc1*, there is also a phase advance, albeit less pronounced, for *CO1* (1.1 h), *CO2* (4.9 h), *LHY2* (1.2 h), and *CDF2* (1.2 h), along with a 1.7-fold increase in the amplitude of *CDF2* (Fig. 2c, d, and Supplementary Fig. 5). Last, in *gi*, we observed a remarkable overexpression of *CDF2* with an amplitude increase of more than 2.1-fold and a phase advance of 7.5 h, coinciding with the peak of *FT2* transcription in the WT (Fig. 2c, d, and Supplementary Fig. 5). These data emphasize the central role for GI in the activation of *FT2*, as well as likely redundant role of LHY2 and CDF2 in repressing *FT2* once its peak of expression is established. Our data also indicate a putative role for TOC1 in repressing *FT2* during the day to complement LHY2 and CDF2 action.

In the current model of FT regulation, TOC1 has never been associated with daytime repression, neither in poplar nor in *Arabidopsis*, even though it is known to bind the common regulator occupied by CO[9–14]. Furthermore, we found that TOC1-CCT domains for DNA binding are conserved between both species, and we identified the putative binding sites for *Arabidopsis* TOC1 and CO present in poplar *FT2* promoters (Supplementary Fig. 6). Based on this evidence, we tested whether transiently overexpressing 35S::TOC1 construct in poplar represses *FT2* transcription. Remarkably, we observed a significant reduction of more than 0.6-fold in the expression of *FT2* under LD, specifically at its peak of expression, with no noticeable change in the transcription of either *CO1* or *CO2* (Fig. 2e). These results indicate that TOC1 represses *FT2* independently of CO1/2 and likely shapes the characteristics of *FT2* expression peak.

### A minimal computer model recapitulates experimental patterns of FT2 transcripts, both under changes in daylength and in gain- or loss-of-function mutants for TOC1, GI, and LHY2

To better understand the interaction between circadian clock elements in *FT2* transcript regulation we constructed an experimental data-driven computational model. This model represents a system of Ordinary Differential Equations (ODEs) that describes relations between changes in transcripts of core clock genes and *FT2* expression (See Supplementary Note 1 for model details). In particular, we intended to construct a minimal quantitative model that yields predictions directly comparable to experimental measurements. Based on experimental findings, we considered that *FT2* upregulation is largely mediated by a combination of GI's direct and indirect activation effects since it is an essential modulator of *FT2* transcription (Fig. 2b). Given the considerable time lag between *GI* and *FT2* expression, we hypothesized a putative daytime repressor of FT2 that would prevent early *FT2* transcription. TOC1 would fulfill this function by competing with the activators, such as CO-like genes, for the binding to the *FT2* promoter. Additionally, we recognized the necessity of a second repressor to control *FT2* expression after its peak time, and our experimental data indicates that LHY2 fits this role by repressing *FT2* during the nighttime and early morning (Fig. 3a).

Our model produces a fair quantitative fit for both the experimental pattern of *FT2* expression in LD and the quantitative downregulation observed after gradually shortening the daylength (Fig. 3b–d). A similar fit was obtained when considering CDF2 instead of LHY2 as the second repressor (Supplementary Fig. 7), suggesting redundancy in downregulating *FT2*. Model simulations also indicate that, under LD, GI phase lags behind TOC1, creating a temporal window bounded earlier by TOC1 and later by LHY2, which boosts *FT2* expression towards the end of the daytime. In SD condition, GI advances its phase while TOC1 remains unchanged, causing them to be expressed simultaneously, which results in a combined inhibitory effect on *FT2* upregulation. This inhibition is further reinforced by the advancement of LHY2 (Fig. 3e). Gene expression pattern changes of clock components interpolated between LD and SD predict a progressive narrowing of the *FT2* activation frame, leading to its gradual downregulation

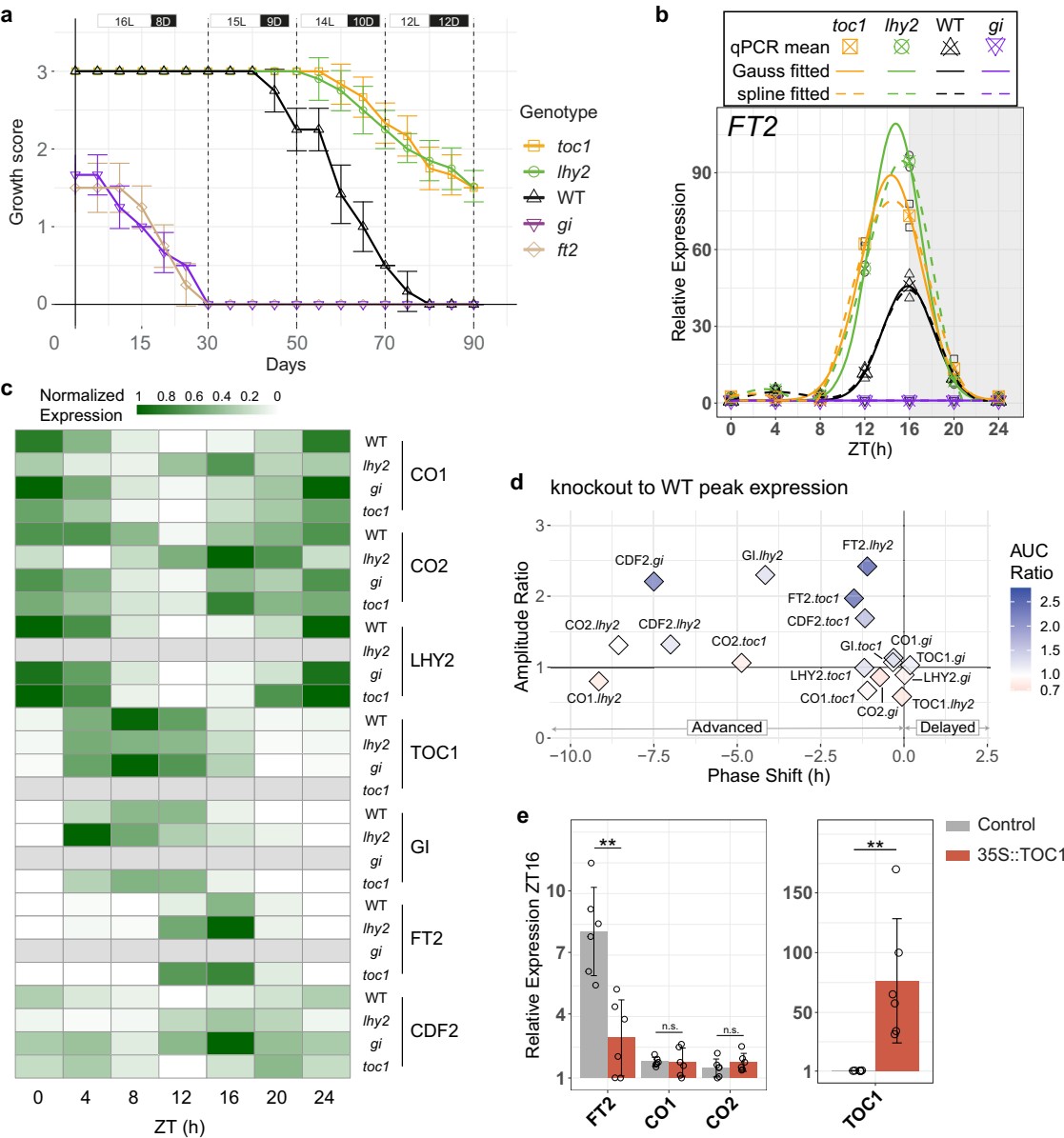

**Fig. 2 | Circadian clock core genes TOC1, GI, and LHY2 regulate FT2 transcription. a** Growth scores from 3 (full growth) to 0 (no growth) in LD and gradually shortening the daylength, indicated by boxes on the top, for WT (black) and loss-of-function lines of TOC1 (yellow), LHY2 (green), GI (purple), and FT2 (beige)[6]. Mean ± sd, $n = 6$ biological replicates. **b** Daily *FT2* transcription in WT (upward triangles, black) compared to the knockout lines *toc1* (squares, yellow), *lhy2* (circles, green), and *gi* (downward triangles, purple) under LD condition. Spots depict qPCR results, with means of the two biological replicates used for fitting highlighted. Solid and dashed lines represent Gaussian and polynomial spline fits, respectively. **c** Heat map of daily transcription patterns for photoperiodic regulators in WT and

*toc1*, *lhy2*, and *gi* knockouts, under LD. qPCR means (**b** and Supplementary Fig. 5) normalized 0-1 per gene. Grey color indicates non-detection levels. **d** Deviation of the gene expression patterns shown in (**c**), comparing the respective knockout line to WT, for phase (time of maximal expression) shift, amplitude ratio, and daily total transcript ratio (AUC−Area Under the Curve). **e** Relative mRNA accumulation of *FT2*, *CO1*, and *CO2* at ZT16 (left panel) after transiently overexpressing 35 S::TOC1 (red) or empty vector used as control (grey) in LD. Right panel shows TOC1 levels. Mean ± sd, $n = 6$ biological replicates. Two-sided Student's *t*-test, p-values from left to right: 0.0012 (**), 0.9558 (n.s.−not significant), 0.3056 (n.s.), and 0.0053 (**). **b**, **c**, **e** ZT, Zeitgeber Time in hours.

transitioning to SD, again consistent with our experimental observations (Fig. 1c and Supplementary Fig. 8; Supplementary Movie 1).

Next, we evaluated the predicted *FT2* expression after gain- or loss-of-function for circadian clock genes in LD regime. As experimentally observed in *gi* knockout, simulations predict complete suppression of *FT2* due to lack of activation (Fig. 3f), whereas GI constitutive overexpression results in a pronounced and widespread increase of *FT2* throughout the entire day, as previously reported (Fig. 3i)[4]. For *toc1* mutant, our model accurately captures the upregulation and phase advance of *FT2* transcription (Fig. 3g). Conversely, simulated TOC1 overexpression downregulates *FT2*, albeit to a higher

extent than experimentally observed after replicating TOC1 ectopic expression level at ZT16 (Fig. 3j−yellow solid line). In fact, a 0.6-fold change for *FT2* expression is achieved with a more moderate overexpression of TOC1 (Fig. 3j−yellow dashed line). Simulated LHY2 overexpression fully suppresses *FT2* transcription (Fig. 3k), while LHY2 increased expression after 4-hour night extension also represses *FT2*, as experimentally reported (Supplementary Fig. 9)[5]. Regarding *lhy2* loss-of-function, it exhibits an upregulation of *FT2* as expected, but earlier than observed (Fig. 3h). This sole discrepancy between model prediction and experimental observations can be attributed to the *FT2* expression definition in the model that strongly relies on GI alone,

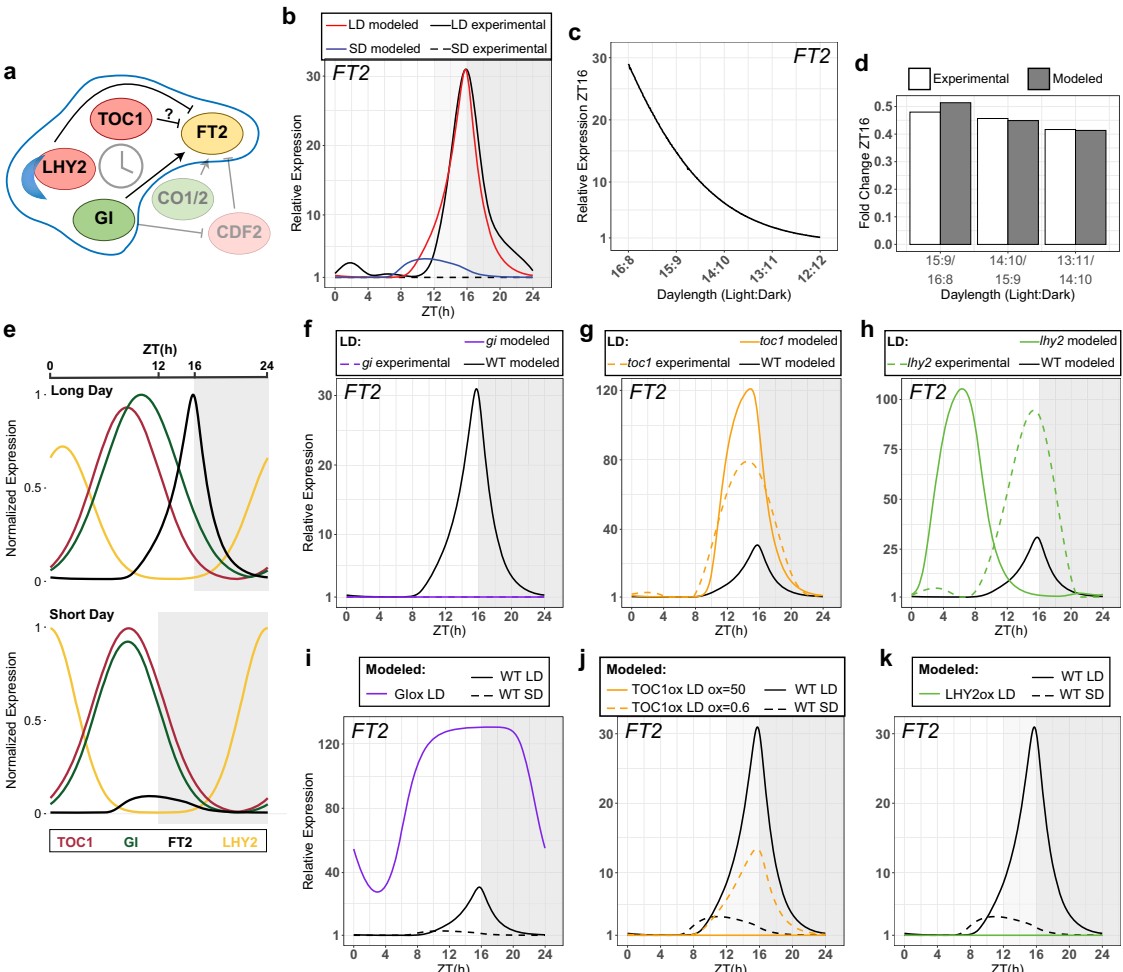

**Fig. 3 | A minimal model that incorporates circadian regulation by TOC1, GI, and LHY2 reflects FT2 transcription dynamics under changing daylengths and genetic variations. a** Photoperiodic regulators and interactions controlling *FT2* expression described in poplar[4,5,21]. TOC1 repressive role is suggested in this work. Circadian clock genes integrated in the model are highlighted. Green and red colors denote positive and negative regulators, respectively. **b** Simulated *FT2* transcription fit (LD-red, SD-blue) to experimental data (spline fitted; LD-black/solid, SD-black/dashed). **c** Simulated *FT2* expression at ZT16 (peak) under different daylengths. **d** Predicted *FT2* expression fold change at ZT16 for indicated photoperiods (values in **c**; grey bars) compared to experimentally observed (calculated from mean values shown in Fig. 1c; white bars). **e** Phase expression patterns of TOC1

(red), GI (green), and LHY2 (yellow) open a window for *FT2* transcription (black) in LD (top panel), which closes transitioning to SD to suppress *FT2* expression (bottom panel). Gene expression is normalized to 0-1 for visualization. Related to Supplementary Fig. 8 and Supplementary Movie 1. **f–h** *FT2* expression predictions in simulated loss-of-function mutants for GI (**f**), TOC1 (**g**), and LHY2 (**h**) (solid lines) compared to experimental data (spline fitted; dashed lines) under LD. WT expression in black. **i–k** Simulated *FT2* transcription after constitutive over-expression of GI (**i**), TOC1 (**j**), and LHY2 (**k**) compared to WT in LD (black/solid) and in SD (black/dashed) predictions. **j** For TOC1, two levels of ectopic expression were considered. See Supplementary Note 1 for details. ZT, Zeitgeber Time in hours.

which is significantly advanced in *lhy2* mutant, coupled with the lack of any repression at that time in the simulation (Fig. 2c, d, and Supplementary Fig. 5), which suggests missing factors mediating GI activation. Identifying additional *FT2* transcriptional regulators, such as CO-like genes that function under natural conditions would aid in quantitatively aligning the predictions for *toc1* and *lhy2* mutants.

Lastly, we conducted a sensitivity analysis of *FT2* transcription deviation following artificial perturbations of the expression patterns of these clock genes used as input, as a proxy for potential disturbances by factors not considered in our study. These simulations confirm the opposing and competitive contribution of GI (promoting) and TOC1 (repressing) in their influence on *FT2*. Additionally, they clarify the role of LHY2 in dampening the *FT2* peak and preventing its expression outside its designated timeframe (Supplementary Fig. 10).

In summary our experimental results and model predictions support that a minimal system composed of the core circadian clock genes LHY2, TOC1, and GI largely explains the key aspects of *FT2* temporal dynamics in response to the photoperiodic switch.

Furthermore, our findings clarify the conserved mechanism for these clock components in FT regulation as well as highlight key differences between *Arabidopsis* and poplar that relates to specific temporal changes in clock gene expression dynamics.

## Discussion

Plant photoperiodic time measurement mechanism relies on the coincidence of daylength and the circadian transcriptional activation of Flowering Locus T[5,8,21,28]. The expression of poplar ortholog FT2 is essential to set the boundaries of the annual vegetative growing season[6,29]. Here, we further demonstrate that *FT2* expression level mediates the growth cessation response quantitatively. The shortening of daylight positively correlates with the decrease in the *FT2* peak amplitude, and these *FT2* levels are inversely linked to the photoperiod sensitivity in the studied genotypes. Specifically, in *ft2* and *gi* mutant lines where *FT2* expression is undetectable, growth ceases even under growth-promoting LD condition. Conversely, in *lhy2* and *toc1* mutants, growth continues under SD, with higher levels of *FT2*

daily expression than WT (Figs. 1c, 2a, and b). This photoperiod sensitivity further underscores the relevance of each circadian clock gene in regulating the daily activation of *FT2*.

This study identifies TOC1 as a novel repressor of *FT2* expression (Fig. 2b and e). In Arabidopsis, *FT* upregulation in the *toc1-1* knockout was associated with a subtle increase in *CO* transcription during the light hours period, when CO protein is stable[8,30]. Likewise, poplar *toc1* mutant displayed a slight phase advance of both *CO1* and *CO2* compared to WT (Fig. 2c, d, Supplementary Fig. 5a, b). However, ectopic TOC1 overexpression, which leads to *FT2* downregulation, did not alter *CO1/2* mRNA levels (Fig. 2e), raising doubts about the regulation of *CO1/2* transcription by TOC1. Based on these findings, we suggest that TOC1 regulates *FT2* transcription directly and propose that it acts through protein-protein competition, interfering with the binding of *FT2* activators to the promoter (Supplementary Fig. 6)[9,10], consistent with model predictions (Fig. 3). Although partially supported by previous reports, the proposed model of TOC1 acting through the NF-YB/C complex would require further investigation to test the validity of this scenario both in *Arabidopsis* and poplar.

Our results confirm that LHY2 negatively impacts poplar active growth, showing a similar effect to TOC1 during growth cessation and as a repressor of *FT2* transcription (Fig. 2a and b). Nightlength extension promptly induces LHY2 to swiftly downregulate *FT2* expression, potentially via 3'UTR interaction[5]. Integrating this direct regulation predicts *FT2* transcription in LD, transitioning to SD (Fig. 3a–e), and after nightlength extension (Supplementary Fig. 9), supporting LHY2 direct repressive role. qPCR patterns in *lhy2* mutant suggest LHY2 may also control *FT2* by modulating the other clock genes. Specifically, we observed an advance of *GI* and a downregulation of *TOC1* in LD (Fig. 2c, d, and Supplementary Fig. 5). *Arabidopsis* orthologs exhibit cross-regulation of LHY over GI and TOC1 that, if conserved in poplar, would explain these changes[29].

GI is indispensable to promote active growth. Our quantitative results prove that *gi* phenocopies *ft2* mutant and both experimental qPCR patterns and model predictions indicate that it is necessary for *FT2* daily expression (Figs. 2a, b, and 3f). Moreover, our findings provide insights into its mode of action. GI was reported to act through the control of CDFs for *FT2* photoperiodic activation[4]. Experimental observations show that *CDF2* mRNA is strongly advanced in *gi* to the time of *FT2* expression, which could account for *FT2* repression (Fig. 2c, d, and Supplementary Fig. 5). In addition, *CDF2* shifts mirror *GI* changes both transitioning to SD and in *lhy2* (Figs. 1e, 2d, Supplementary Figs. 2, and 5)[4,19]. In *Arabidopsis*, GI represses CDFs through protein degradation mediated by FKF1[17]. Here, we show that GI affects *CDF2* transcription, a link not covered in *Arabidopsis*. Furthermore, the absence of transcriptional response of *FKF1* in poplar to changes in daylength, unlike in Arabidopsis where *FKF1* phase advances under SD, suggests a minor role of FKF1 in *FT2* photoperiodic control (Fig. 1d, e, Supplementary Figs. 2, and 3).

Beyond CDFs regulation, the absence of *FT2* expression in *gi* knockout (Fig. 2b) suggests that GI also plays a key role in promoting *FT2* activation, modeled as a direct effect in our work (Fig. 3a). However, assuming that *FT2* activation solely relies on GI results in an early induction in *lhy2* mutant simulation (Fig. 3h), which is inconsistent with experimental observations (Fig. 2b). This implies the involvement of yet unknown downstream or interacting factors in mediating GI activation. Indeed, there is no evidence that GI functions as a transcription factor despite its association with specific DNA targets[4,19,31]. A priori, we might consider GI binding mediated by CO closest orthologs. Nonetheless, *CO1/2* expression patterns do not explain *FT2* activation in LD in poplar WT, as they are mainly transcribed at night, when the proteins are presumably non-functional (Supplementary Fig. 2a, b)[20]. Additionally, we found that transcriptional control of *CO1/2* differs from that of *CO* in *Arabidopsis*. The *gi* loss-of-function does not impact on *CO1* nor *CO2* expression in LD, which is particularly striking given the strong

upregulation of *CDF2* that would lead to *CO1/2* repression[4]. Moreover, *lhy2* knockout also advances *CO1* and *CO2* phases (Fig. 2c, d, and Supplementary Fig. 5). In *Arabidopsis*, loss of LHY function leads to an analogous increase in *FT* expression but without affecting *CO* transcription, elucidating *FT* upregulation through CO protein stabilization[32]. These data suggest that there is an additional link between LHY2 and *CO1/2* in poplar. Overall, this divergence in *CO1/2* regulation may explain their limited impact on *FT2* activation, with CO-independent pathways governed by GI gaining higher relevance. Poplar *co1/2* knockout would confirm or refute their necessity for *FT2* expression. Consequently, other CO-like proteins with conserved functional domains may mediate GI activity with more consistent expression times for *FT2* activation[3]. Additionally, these could directly compete with TOC1 to shape *FT2* expression. Furthermore, we cannot dismiss the participation of genes from other families, such as Gbox-binding transcription factors given the enrichment of this motif in GI targets in *Arabidopsis*[31], or the potential additive effect of multiple factors, which opens up new avenues for study.

The combined temporal expression dynamics of the circadian clock genes TOC1, GI, and LHY2 in response to daylength accurately explain the changes in *FT2* transcription. This study establishes a minimal model comprising three components-one activator and two repressors- in which the phase shifts controlled by the photoperiod play the pivotal role. Through computational modeling, we demonstrate the viability of GI integrating the pathways that induce *FT2* under LD condition. As a result, GI creates a specific time window that enables *FT2* expression in LD. During the transition to SD, this window is closed earlier in the daytime by TOC1 (via GI phase advance) and later by LHY2. Assuming that both GI and LHY2 phases progressively shift as daylength shortens, we infer that under natural conditions, from the beginning of summer, the expression window of *FT2* gradually narrows, leading to a continuous reduction in *FT2* expression level. Eventually, FT2 reaches a lower threshold initiating growth cessation in autumn (Fig. 3e and Supplementary Fig. 8; Supplementary Movie 1).

Our model exemplifies an internal coincidence clock-driven mechanism in which core clock genes control the *FT2* activity window. However, clock-independent external cues could further contribute to the modulation of FT2 expression. For instance, a rapid lengthening of night extends LHY2 expression, thereby repressing *FT2* (Supplementary Fig. 9)[5]. In addition, putative factors mediating GI activation could integrate an external coincidence mechanism akin to CO in *Arabidopsis*, although, as proven in this work, not necessarily. Nevertheless, our study captures the key differences and similarities between *Arabidopsis* and poplar. In *Arabidopsis*, GI precedes TOC1 expression regardless of daylength, which would block *FT* upregulation in LD as modeled here, indicating the requirement of CO-mediated activation. The increased expression of GI in SD would also contradict *FT* downregulation (Supplementary Fig. 3). By contrast, in poplar TOC1 is advanced with respect to GI allowing for GI-dependent *FT2* upregulation. Alternatively, in SD condition, both TOC1 and GI overlap in phase leading to strong downregulation of *FT2* transcripts (Fig. 1d).

This study allowed us to identify the necessary circadian regulators and interactions for daylength control of the photoperiodic effector FT2. Future research, including explicit modeling of light input and more detailed description of the interactions presented, would contribute to build a more comprehensive framework that directly link *FT2* to the environmental influences. This could lead to new strategies for modulating *FT2* expression to improve tree geographical adaptation and, therefore, plantation forest breeding.

## Methods

### Plant material and growth conditions
Hybrid poplar *Populus tremula x alba* INRA clone 717 1B4 was used as *wildtype* for gene expression assays and plant transformation. Poplar

plantlets were cultivated in vitro in Murashige and Skoog (MS) medium 1B (pH 5.7), supplemented with 2% sucrose, indole acetic and indole butyric acids (0.5 mg/L), and 0.7% (w/v) plant agar. Plants were grown under 16 h light:8 h dark photoperiod, 21 °C, 65% humidity, and 300-350 μmol m-2 s-1 PPFD (Photosynthetic Photon Flux Density) conditions for 2 weeks. For time-course qPCR experiments, poplars were kept in vitro for an additional week under the same conditions in LD assays. Alternatively, for shorter daylengths, photoperiod was adjusted accordingly before use. For night extension experiment, poplars grown in LD were subjected to a 4-hour dark period extension.

### Generation of CRISPR-Cas9 stable lines

To generate CRISPR-Cas9 constructs targeting TOC1 or GI within p201N-Cas9 plasmid, specific (close to 5′-end) single guide RNA (sgRNA) were chosen from a pre-designed SNP-free dataset available on AspenDB (Supplementary Fig. 4)[6,33]. *Agrobacterium tumefaciens* strain GV3101/pMP90, carrying the p201N-Cas9 vector with the appropriate sgRNA, was used to transform hybrid poplar. Positive transformed poplar explants were regenerated into new plantlets[6]. Genome edition was assessed by sequencing after amplifying the flanking sgRNA site using specific primers (Supplementary Table 1). Resulting sequences were aligned using ClustalW multiple alignment tool in the BioEdit Sequence Alignment Editor 7.0 to identify the predicted protein truncation, as indicated in Supplementary Fig. 4[6]. Only lines showing non-functional protein predictions for both *tremula* and *alba* haplotypes were considered as positive mutants.

### Transient overexpression

To create the TOC1 overexpressing construct, TOC1 transcript was amplified from hybrid poplar genome using the primers TOC1_FW 5′-ATGGAGGGAGAGGTAGATGAGC-3′ and TOC1_RV 5′-TTAAGATCCTG AAGCATCGTCCTCAG-3′. The resulting piece was cloned along with 35 S promoter into dpGreen destination vector using the MultiSite Gateway Kit (Invitrogen, MA, United States). Empty construct used as control was assembled in the same manner without TOC1 sequence. In vitro poplar plantlets were transformed following the protocol previously reported[34], and leaf samples were collected after 2 days at the peak of *FT2* expression.

### Plant phenotyping

In vitro poplars were transplanted to 3.5 L pots filled with blond peat at pH 4.5 and kept under same LD growth condition. Phenotyping for the selected lines was initiated once WT plants had reached full active growth. Daylength was gradually shortened to evaluate growth cessation progression, scored from 3 (full growth) to 0 (growth absent and apical bud formed)[35].

### RT-qPCR expression analysis

Total RNA was extracted from young leaves of poplar plantlets using NucleoSpin RNA Plant kit (Macherey-Nagel, Düren, Germany). First-strand complementary DNA (cDNA) was synthesized using Maxima First Strand cDNA Synthesis Kit with dsDNase (Thermo Fisher Scientific, MA, United States). Quantitative real-time PCR (qPCR) analyses were carried out in a Roche LightCycler 480 II instrument (Roche Diagnostics, Barcelona, España), and values were obtained using the relative quantification method[36]. Results were relativized to UBQ7[37], and the log2 fold change was calculated by setting 1 the minimum expression for each gene. A list of the primers used for qPCR analysis is provided in Supplementary Table 1. For the time course experiments qPCR data generated is provided in Supplementary Data 1.

### Accession numbers

Sequences of the genes called in this study can be found in Phytozome13 database with next identifiers. For poplar: *CDF2* (PtXaTreH.08G068800; PtXaAlbH.08G072200), *CO1* (PtXaTreH.17G0 90200; PtXaAlbH.17G081800), *CO2* (PtXaTreH.04G088300; PtXaAlbH.04G087300), *FKF1* (PtXaTreH.10G086700; PtXaAlbH.10G082000), *FT2* (PtXaTreH.10G148500; PtXaTreH.10G148700; PtXaAlbH.10G14 2000 PtXaAlbH.10G142200), *GI* (PtXaTreH.05G148500; PtXaAlbH.05 G150700), *LHY2* (PtXaTreH.14G082800; PtXaAlbH.14G083900), and *TOC1* (PtXaTreH.15G047900; PtXaAlbH.15G047900). There are 2 copies of *FT2* in each haplotype derived from a local duplication[38]. For *Arabidopsis*: *CDF2* (AT5G39660), *CO* (AT5G15840), *FKF1* (AT1G68050), *FT* (AT1G65480), *GI* (AT1G22770), *LHY* (AT1G01060), and *TOC1* (AT5G61380).

### Data-driven model of FT2 transcription

The computer model was built on MATLAB_R2022a (Mathworks, Cambridge, UK). Transcriptional daily patterns of *TOC1*, *GI*, *LHY2*, and *CDF2* were used as input to model the expression of *FT2* under simulated light:dark cycles and mutant lines. Input patterns were fitted to a Gaussian pulse dependent of daylength. *FT2* expression was defined by an Ordinary Differential Equation (ODE). A more detailed description of the methods used is provided in Supplementary Note 1. Optimal parameters values are listed in Supplementary Table 2. MATLAB code that allows the simulation of the model under different daylengths and genotypes scenarios can be found in Supplementary Data 2.

### Reporting summary

Further information on research design is available in the Nature Portfolio Reporting Summary linked to this article.

## Data availability

All data supporting the findings of this study are available within the article and its Supplementary Information. Poplar and *Arabidopsis* accession numbers are indicated in Methods section. Raw values for *Arabidopsis* daily patterns of transcription (Supplementary Fig. 3) were obtained from the microarray datasets "long-day" and "short-day" available in DIURNAL database from Mockler Lab (http://diurnal. mocklerlab.org/diurnal_data_finders/new). Source data are provided with this paper.

## Code availability

The full MATLAB code that runs the model of *FT2* transcription is provided in separate Supplementary Data 2. Refer to README file for more information. The code can also be found on https://github.com/dalique1996/FT2-Expression-Model.

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

## Acknowledgements

This work was supported by Ministerio de Ciencia Innovacion y Universidades of Spain (PGC2018-093922-B-I00 and PID2021- 123060OB-I00 to IA and MP; PID2021-122158NB-I00 to KW), Programa de Atraccion de Talento 2017 (Comunidad de Madrid 2021-5 A/BIO-20952 to KW), and Severo Ochoa (SO) Program for Centres of Excellence in R&D from the Agencia Estatal de Investigación of Spain [grant CEX2020-000999-S (2022 to 2025) to the CBGP]. DA was funded by the Ministerio de Universidades of Spain FPU19/04729 fellowship. NGS is supported by the Maria Zambrano grant (UP2021-035).

## Author contributions

D.A., I.A., K.W., and M.P. conceived the project. D.A. performed most of the experiments. A.R.L. generated the CRISPR-lines. N.G.S. performed the transactivation assays. D.A. and K.W. designed and performed the modeling and simulations. D.A., K.W., and M.P. wrote the manuscript.

## Competing interests

The authors declare no competing interests.
