## [Peer Review File · Nature Communications]

Core Clock Genes Adjust Growth Cessation Time to Day-Night Switches in Poplar.Reviewer #1 (Remarks to the Author):

The manuscript by Alique et al. is a nice contribution to our understanding of how photoperiod modulates seasonal growth in trees. The authors provide a minimal model that attempts to explain the effect of photoperiod on growth through the interactions of a set of three genes that are associated with circadian clock function and, apparently, directly control the expression of FT2, the key gene controlling growth in response to photoperiod in poplar. The results are based on a combination of gene expression analysis in wild type trees as well as in trees in which key clock associated genes have been edited through crispr, namely TOC1, LHY and GI. The results of the physiological and molecular analysis are combined with a minimal mathematical model that attempts to explain photoperiodic control of FT2 mRNA production based on positive effects of GI and FT2 expression and direct negative effects of LHY and TOC1, in combination with the effects of these clock genes on the photoperiodic dependent phase of expression of several other genes controlled by the clock and linked to the regulation of FT2 expression.

The manuscript is well written, the methods are sound and the hypothesis/model proposed is interesting and exciting.

My major concern is that, if I understood correctly, the model assumes that photoperiod affects FT2 expression only indirectly, through its effect on clock entrainment, but not through additional direct regulation of levels or activity of some factor controlling FT2 expression. If this were the case, the model would be based on an internal coincidence mechanisms, in which only under certain photoperiods the phase of expression of FT2 mRNA regulators are such that FT expression is induced or repress, but there is no direct effect of a photosensory pathway on the direct regulation of FT2 expression. While this is possible, I would like the authors to explicitly state this and propose (not necessarily perform for this manuscript) experiments to discriminate this model from an external coincidence model such as is the case for the photoperiodic regulation of flowering in plants such as Arabidopsis. One possibility would be to evaluate the effect of long versus short thermoperiods on growth under continuous light conditions, similarly to what was done in the 70's in insects (Thermoperiodic Control of Diapause in an Insect: Theory of Internal Coincidence, SCIENCE 27 Jul 1973, Vol 181, Issue 4097, pp. 358-360).

The second issue is that the authors are proposing that TOC1 is a direct regulator of FT2 expression, but this is only based on the effect of mutations or overexpression of TOC1 on FT2 mRNA. It would be interesting to evaluate whether TOC1 directly binds genomic regions known to control FT2 expression, similarly to what the authors did with LHY previously. My guess is that this may not be simple, as TOC1 probably does not bind DNA directly but only through its interaction with other factors, which may complicate the analysis of binding through DAP-seq, as done before by the group of Mariano Perales. In any case, I would like the authors to mention these limitations and indicate what would be required here or in the future to provide more compelling evidence of direct TOC1 regulation of FT2 expression.

All in all I like the manuscript and its implications, it is an exciting and provocative idea. Would benefit from additional evidence, but not necessarily for this manuscript.

Reviewer #2 (Remarks to the Author):

To the Authors:

This manuscript seeks to elucidate how the circadian clock genes regulate the timing of Growth Cessation in poplar. The authors utilized mutants of key genes and CRISPR/Cas9 loss-of-function lines to control and quantify the expression level of FT2. Through the analysis of phase and amplitude data of expression rhythms under long-day (LD) and short-day (SD) conditions, the authors attempt to uncover the network that regulates FT2 in poplar. These findings are significant and noteworthy in the field of circadian rhythms. I particularly believe that the mathematical model of this study successfully quantifies the relationships between genes.

For the full potential of this research to be realized, it is essential to validate and ensure the accuracy of the manuscript's mathematical model. The following questions should be addressed:

(1) In Figure 1D and Figure 2C, what is the rationale behind the order of the genes (LHY2, GI, TOC1, ...)? Are they sorted by their peak times? Understanding this arrangement might help in

comprehending the circadian clock's role in Growth Adjustment.

(2) Regarding Figure 3G, is there a need to reconcile the discrepancy between the experiment and the model? While the manuscript discusses potential causes, I feel that the inconsistency in the relationship between LHY and FT2 should be more thoroughly addressed.

(3) In the lower panel of Figure 3D, the peak expression of FT2 appears to be delayed by about 3 hours after the onset of the dark period. Is this delay consistent with the referenced Arabidopsis model (D. D. Seaton et al., 2015)? In the Arabidopsis mathematical model, does the expression of FT decrease immediately following the onset of the dark period? Clarifications on the differences between the referenced Arabidopsis model and the presented poplar model are essential. For example, understanding characteristics such as whether light inputs directly influence the circadian clock model can help ensure accurate discussions about phase shifts in circadian rhythms.

(4) While Figure S1 presents the computational results under LD conditions, it might be beneficial to also include results under SD conditions. Incorporating these results to compare and contrast behaviors between Arabidopsis and poplar would enhance the discussion.

(5) In Figure 1D, FT2 expression under SD conditions hasn't been experimentally measured. Yet, the mathematical model anticipates a measurable expression level (FT2 expression level is about 0.2 at ZT=13~14 h in Figure 3D). Is it feasible to experimentally measure this? If FT2 is genuinely not expressed and is unmeasurable, could it be necessary to revise the model?

(6) The idea of "a window for FT2 transcription" is intriguing. Yet, this window isn't clearly defined in the data (the start and end times of the window remain unclear). Could you explore the daylength dependence of this window through simulations and possibly present it as supplemental material?

(7) Can the minimal data-driven model introduced in this manuscript predict or explain results under untrained daylength conditions, like Light:Dark=15:9, 14:10, or 13:11? Such verification would help confirm the model's accuracy in representing the circadian clock's regulation of FT for growth adjustment.

Minor Comments:

Supplementary Movie: Displaying the daylength value in the animation would be beneficial.

Description for Supplementary equation (2): Should "a" and "r" be denoted as "a" and "r₁"? Specifying subscripts meticulously would be advisable.

We want to thank two Reviewers for their input and suggestions that helped us to improve the manuscript. In particular, we have re-parametrized the model (Table S2) to improve the fit of *FT2* downregulation transitioning to SD, as observed in our experimental data. Accordingly, Fig. 3 and Supplementary Fig. 7 have been updated with new simulations. Additionally, concerning Fig. 2E, we have increased number of biological replicates (from n=4 to n=6) and included statistical significance. Details of changes and improvements over the original version of the manuscript have been included below in Response to Reviewers.

Reviewer #1

The manuscript by Alique et al. is a nice contribution to our understanding of how photoperiod modulates seasonal growth in trees. The authors provide a minimal model that attempts to explain the effect of photoperiod on growth through the interactions of a set of three genes that are associated with circadian clock function and, apparently, directly control the expression of *FT2*, the key gene controlling growth in response to photoperiod in poplar. The results are based on a combination of gene expression analysis in wild type trees as well as in trees in which key clock associated genes have been edited through Crispr, namely *TOC1*, *LHY* and *GI*. The results of the physiological and molecular analysis are combined with a minimal mathematical model that attempts to explain photoperiodic control of *FT2* mRNA production based on positive effects of *GI* and *FT2* expression and direct negative effects of *LHY* and *TOC1*, in combination with the effects of these clock genes on the photoperiodic dependent phase of expression of several other genes controlled by the clock and linked to the regulation of *FT2* expression. The manuscript is well written, the methods are sound, and the hypothesis/model proposed is interesting and exciting. My major concern is that, if I understood correctly, the model assumes that photoperiod affects *FT2* expression only indirectly, through its effect on clock entrainment, but not through additional direct regulation of levels or activity of some factor controlling *FT2* expression. If this were the case, the model would be based on an internal coincidence mechanism, in which only under certain photoperiods the phase of expression of *FT2* mRNA regulators are such that *FT* expression is induced or repress, but there is no direct effect of a photosensory pathway on the direct regulation of *FT2* expression. While this is possible, I would like the authors to explicitly state this and propose (not necessarily perform for this manuscript) experiments to discriminate this model from an external coincidence model such as is the case for the photoperiodic regulation of flowering in plants such as *Arabidopsis*. One possibility would be to evaluate the effect of long versus short thermoperiods on growth under continuous light conditions, similarly to what was done in the 70's in insects (Thermoperiodic Control of Diapause in an Insect: Theory of Internal Coincidence, *SCIENCE* 27 Jul 1973, Vol 181, Issue 4097, pp. 358-360).

Author's Reply: Thank you for these constructive comments. As correctly indicated, our minimal computational model depends on an internal coincidence clock-driven mechanism for *FT2* regulation, which is directly modulated by entrained oscillations of circadian clock genes. Particularly, our model integrates experimentally observed *LHY2*, *TOC1*, and *GI* patterns as inputs to predict dynamics in *FT2* transcription. We agree there

is evidence that external cues, such as light, also influence FT2 expression, although it has not been the focus of this work. For instance, in night extension experiments FT2 is downregulated. This is linked to the enhanced activity of LHY2 repressor (Ramos-Sánchez et al., 2019). Accordingly, our model predicts FT2 downregulation by integrating the increase of LHY2 expression after a 4-hour night extension that is in agreement with that observed experimentally (Ramos-Sánchez et al., 2019). We added new simulations in Supplementary Fig. 9 (Lines 225-226). Furthermore, we have discussed this aspect and shortcomings of the current model in the revised version of the manuscript (Lines 333-346).

Ramos-Sánchez, Jose M., et al. "LHY2 integrates night-length information to determine timing of poplar photoperiodic growth." *Current Biology* 29.14 (2019): 2402-2406.

The second issue is that the authors are proposing that TOC1 is a direct regulator of FT2 expression, but this is only based on the effect of mutations or overexpression of TOC1 on FT2 mRNA. It would be interesting to evaluate whether TOC1 directly binds genomic regions known to control FT2 expression, similarly to what the authors did with LHY previously. My guess is that this may not be simple, as TOC1 probably does not bind DNA directly but only through its interaction with other factors, which may complicate the analysis of binding through DAP-seq, as done before by the group of Mariano Perales. In any case, I would like the authors to mention these limitations and indicate what would be required here or in the future to provide more compelling evidence of direct TOC1 regulation of FT2 expression. All in all, I like the manuscript and its implications, it is an exciting and provocative idea. Would benefit from additional evidence, but not necessarily for this manuscript.

Author's Reply: We agree with the Reviewer that explaining the molecular mechanism by which TOC1 represses FT2 transcription would be an exciting addition as a follow-up to this study. TOC1, and the other PSEUDO RESPONSE REGULATORS contain a CCT domain that binds the TGTG cis-element *in vitro* by EMSA assay in *Arabidopsis* (Gendron et al., 2012). Indeed, directed mutation of the CCT domain reduces TOC1 repressor activity (Gendron et al., 2012). In parallel, TOC1 binding is stabilized by forming a trimer with NF-YB/C subunits (Yang et al., 2021), facilitating non-specific chromatin accessibility. The canonical NF-Y complex includes a third component, NF-YA, that confers DNA binding specificity (Nardini et al., 2013). There is a lack of consensus on whether TOC1-CCT domain binds to DNA alone or requires the interaction with NF-Y complex. This question has been further explored for the analogous CCT domain of CO. CO-CCT was reported to bind FT promoter (Tiwari et al., 2010). However, different *nf-y* mutants phenocopy *co* and advanced flowering of 35S::CO is drastically reduced in these *nf-y* backgrounds (Kunimoto et al., 2010; Tiwari et al., 2010); indicating that CO activation of FT transcription *in vivo* requires interaction with NF-Y complex (Cao et al., 2014). The most updated revision suggests that CO may compete with NF-YA for NF-

YB/C interaction for transcriptional regulation, given the similarity between CO-CCT and NF-YA (Gnesutta et al., 2017). These observations presumably extend to the rest of CCT proteins like TOC1, supporting TOC1 competition with *FT2* activators in our computational model. Additionally, TOC1 and CO were reported to be associated *in vivo* (Hayama et al., 2017).

Based on these findings in *Arabidopsis*, a DAP-seq study in poplar using TOC1 as bait would likely target *FT2* promoter since it contains TGTG sites, and the TOC1-CCT is highly conserved between both species (new Supplementary Fig. 6; Lines 174-177). Nevertheless, considering the arguments presented, we think TOC1 binding to *FT2* promoter should be studied in the context of the TOC1/NF-Y heterocomplex. Exploring this angle first requires the identification of poplar homologs and functional characterization of NF-YA/B/C genes, which will take a few years. Alternatively to DAP-seq, finding *in vivo* TOC1 targets using ChIP-Seq will be very informative. To carry out this study, we must complement our poplar *toc1* knockout with a genomic version of 3xFlag-TOC1 fusion for immunoprecipitation. However, cloning poplar TOC1 genomic DNA is challenging since promoter sequences are not yet well annotated, which is a limiting factor that we will address in our future work.

We have clarified this in the manuscript and stated the necessity of forthcoming studies to validate TOC1 mechanism of action (Lines 264-266).

Gendron, Joshua M., et al. "Arabidopsis circadian clock protein, TOC1, is a DNA-binding transcription factor." *Proceedings of the National Academy of Sciences* 109.8 (2012): 3167-3172.

Gnesutta, Nerina, et al. "CONSTANS imparts DNA sequence specificity to the histone fold NF-YB/NF-YC dimer." *The Plant Cell* 29.6 (2017): 1516-1532.

Hayama, Ryosuke, et al. "PSEUDO RESPONSE REGULATORS stabilize CONSTANS protein to promote flowering in response to day length." *The EMBO journal* 36.7 (2017): 904-918.

Kumimoto, Roderick W., et al. "NF-YC3, NF-YC4 and NF-YC9 are required for CONSTANS-mediated, photoperiod-dependent flowering in *Arabidopsis thaliana*." *The Plant Journal* 63.3 (2010): 379-391.

Nardini, Marco, et al. "Sequence-specific transcription factor NF-Y displays histone-like DNA binding and H2B-like ubiquitination." *Cell* 152.1 (2013): 132-143.

Tiwari, Shiv B., et al. "The flowering time regulator CONSTANS is recruited to the FLOWERING LOCUS T promoter via a unique cis-element." *New Phytologist* 187.1 (2010): 57-66.

Yan, Jiawei, et al. "TOC1 clock protein phosphorylation controls complex formation with NF-YB/C to repress hypocotyl growth." *The EMBO Journal* 40.24 (2021): e108684.

Reviewer #2

This manuscript seeks to elucidate how the circadian clock genes regulate the timing of Growth Cessation in poplar. The authors utilized mutants of key genes and CRISPR/Cas9 loss-of-function lines to control and quantify the expression level of FT2. Through the analysis of phase and amplitude data of expression rhythms under long-day (LD) and short-day (SD) conditions, the authors attempt to uncover the network that regulates FT2 in poplar. These findings are significant and noteworthy in the field of circadian rhythms. I particularly believe that the mathematical model of this study successfully quantifies the relationships between genes. For the full potential of this research to be realized, it is essential to validate and ensure the accuracy of the manuscript's mathematical model. The following questions should be addressed:

(1) In Figure 1D and Figure 2C, what is the rationale behind the order of the genes (LHY2, GI, TOC1, ...)? Are they sorted by their peak times? Understanding this arrangement might help in comprehending the circadian clock's role in Growth Adjustment.

Authors' Reply: Thanks for your suggestion. For clarity, we have now sorted them according to their phase (time at which maximal gene expression is observed) in LD. At the beginning of the light period, in poplar it follows (Figs. 1D, 2C, Supplementary Figs. 2, 5): CO1, CO2, LHY2, TOC1, GI, FKF1, FT2, and CDF2, and in *Arabidopsis* (Supplementary Fig. 3A): LHY, CDF2, GI, FKF1, TOC1, FT and CO. This analysis already highlights some key differences between *Arabidopsis* and poplar, such as TOC1-GI interchanged relative phase (see revised version, Lines 134-138) and delayed CO1/2 expression compared to CO (Lines 138-141).

(2) Regarding Figure 3G, is there a need to reconcile the discrepancy between the experiment and the model? While the manuscript discusses potential causes, I feel that the inconsistency in the relationship between LHY and FT2 should be more thoroughly addressed.

Author's Reply: We agree with the Reviewer that the model has room for improvement, as there are missing components in poplar that would regulate FT2 transcription in the absence of LHY2. This study uncovers the fundamental links between core clock genes and FT2. Thus, in our model, GI has a dominant effect on FT2 activation. However, the simulated pattern of FT2 in *lhy2* suggests the presence other downstream or interacting factors mediating GI activity. Following *Arabidopsis* studies, we initially considered CO orthologs for this role. Indeed, defining CO1, instead of GI, as FT2 activator reproduces FT2 upregulation in simulated *lhy2*. However, this activation mediated by CO1 does not explain FT2 expression in LD in the WT scenario, as CO1 is transcribed at night when the protein is presumably non-functional due to its similarity to *Arabidopsis* CO (Valverde, 2004). See the simulated results in the figure attached below. Comparable predictions would be drawn from considering CO2 as activator given CO1-CO2 similar expression patterns (Figs. 1D, 2C, and Supplementary Figs. 2A, 2B, 5A, 5B). It is possible that

CO1/2 or other CO-like genes and GI control FT2 expression in poplar together to compensate for changes in context-dependent repressor activities. Generating *co1/2* knockout in poplar could help to confirm or refute their necessity for FT2 expression. However, weaker CO1/2 RNAi phenotypes compared to GI manipulations point to the implication of additional yet unknown factors (Böhlenius et al., 2006; Ding et al., 2018; Fig. 2B). Other CO-like proteins with conserved functional domains may also be at play, with more consistent expression times for FT2 activation (Triozzi et al., 2018). We have further elaborated in the revised discussion about this discrepancy (Lines 292-302, 312-319).

Figure. For the CO1-activator model, (A) we assumed that CO1 protein (dotted lines) follows the mRNA pattern (solid lines) during the daytime but is rapidly degraded at night. (B) In simulated *lhy2* (solid-green), FT2 expression closely matches the experimental result (dashed-green). However, in simulated WT, FT2 is strongly downregulated (solid-black) compared to the upregulation observed experimentally (dashed-black). In simulated WT, CO protein is expressed only at the beginning of the day (A), when LHY2 and TOC1 repressions counteract its activation role.

Böhlenius, Henrik, et al. "CO/FT regulatory module controls timing of flowering and seasonal growth cessation in trees." *Science* 312.5776 (2006): 1040-1043.

Ding, Jihua, et al. "GIGANTEA-like genes control seasonal growth cessation in Populus." *New Phytologist* 218.4 (2018): 1491-1503.

Triozzi, Paolo M., et al. "Photoperiodic regulation of shoot apical growth in poplar." *Frontiers in plant science* 9 (2018): 1030.

Valverde, Federico, et al. "Photoreceptor regulation of CONSTANS protein in photoperiodic flowering." *Science* 303.5660 (2004): 1003-1006.

(3) In the lower panel of Figure 3D, the peak expression of FT2 appears to be delayed by about 3 hours after the onset of the dark period. Is this delay consistent with the referenced Arabidopsis model (D. D. Seaton et al., 2015)? In the Arabidopsis mathematical model, does the expression of FT decrease immediately following the onset of the dark period? Clarifications on the

differences between the referenced Arabidopsis model and the presented poplar model are essential. For example, understanding characteristics such as whether light inputs directly influence the circadian clock model can help ensure accurate discussions about phase shifts in circadian rhythms.

Authors' Reply: Thank you for this comment. In Seaton et al., 2015, *FT* peaks at the light-dark interphase irrespective of daylength, since *FT* transcription directly follows the CO protein level, which is fully suppressed at night (see figure below). Our model does not exhibit such a sharp shift due to the absence of direct night control of *FT2* regulators (Fig. 3E and new Supplementary Fig. 8). This has been clarified in Discussion (see also response to Reviewer #1, Lines 333-339). In poplar, *FT2* expression in SD was undetectable by qPCR, making it unclear whether a specific pattern in SD exists. In any case, this does not impact our findings regarding clock regulation on *FT2*. Improved sensitivity, enabling *FT2* detection in SD, could contribute to assessing the potential implication of intermediary factors, such as CO orthologs, whose stability relies directly on photoperiod.

Figure. Example of simulated CO protein and *FT* mRNA for LD (16h light: 8h dark) and SD (12h light: 12h dark) in Seaton et al., 2015.

Seaton, Daniel D., et al. "Linked circadian outputs control elongation growth and flowering in response to photoperiod and temperature." *Molecular systems biology* 11.1 (2015): 776.

(4) While Figure S1 presents the computational results under LD conditions, it might be beneficial to also include results under SD conditions. Incorporating these results to compare and contrast behaviors between Arabidopsis and poplar would enhance the discussion.

Authors' Reply: Thank you for your suggestion. We have included SD results from Seaton et al., 2015 in Supplementary Fig. 1. *FT* expression changes in SD after LHY2,

TOC1, and GI manipulations exhibit the same trends as in LD. However overall values vary due to lower *FT* transcription in the WT itself.

Seaton, Daniel D., et al. "Linked circadian outputs control elongation growth and flowering in response to photoperiod and temperature." *Molecular systems biology* 11.1 (2015): 776.

(5) In Figure 1D, *FT2* expression under SD conditions hasn't been experimentally measured. Yet, the mathematical model anticipates a measurable expression level (*FT2* expression level is about 0.2 at ZT=13~14 h in Figure 3D). Is it feasible to experimentally measure this? If *FT2* is genuinely not expressed and is unmeasurable, could it be necessary to revise the model?

Author's Reply: Since *FT2* expression is not detectable with our qPCR experimental setup in SD (12h dark: 12h light), we adjusted our model to the downregulation observed at ZT16 from a 16h light: 11h dark to a 13h light: 11h dark scenario (Fig. 1C). We have refined model parametrization by increasing the weight of *FT2* downregulation in the cost function (see new equation 3 in Supplementary Information). New simulations predict a very close fit to the *FT2* fold change experienced transitioning to SD (new Fig. 3D). New parameter values mainly differ in the increased hill coefficients for GI and LHY2 ($n_{1/2}=3$; Table S2). In the updated version, *FT2* is still predicted to be expressed at a much lower value in SD; however, this level may be below the sensitivity of the qPCR technique.

(6) The idea of "a window for *FT2* transcription" is intriguing. Yet, this window isn't clearly defined in the data (the start and end times of the window remain unclear). Could you explore the daylength dependence of this window through simulations and possibly present it as supplemental material?

Author's Reply: Thanks for pointing this out. To clarify the window for *FT2* activation, we have added new Supplementary Fig. 8 that includes the peak area and width with start and end time points. This complements the amplitude at ZT16 used to fit the model shown in Fig. 3C.

(7) Can the minimal data-driven model introduced in this manuscript predict or explain results under untrained daylength conditions, like Light:Dark=15:9, 14:10, or 13:11? Such verification would help confirm the model's accuracy in representing the circadian clock's regulation of FT for growth adjustment.

Author's Reply: Thanks for your suggestion. We built a data-driven model based on experimental transcript patterns of core clock genes. However, we did not explicitly model photoperiod input, as we focused on understanding *FT2* as an outcome of interpreting circadian clock oscillations under natural conditions (LD, SD), rather than focusing on the direct light-dependent control of *FT2* regulators (as discussed with

Reviewer #1). Without light input, untrained conditions cannot be tested. Of course, this is an intriguing prospect for future work to develop a more comprehensive framework that directly links *FT2* to various environmental cues. However, the main focus of this work is how clock genes changes modulate *FT2* expression in the natural entrained context. We have discussed future model extension to include light input and untrained conditions in the discussion (Lines 348-351).

Minor Comments:

-Supplementary Movie: Displaying the daylength value in the animation would be beneficial.

Author's Reply: Thank you for this suggestion. We have included day-night length values within the graph.

-Description for Supplementary equation (2): Should "a" and "r" be denoted as "a" and "r₁"? Specifying subscripts meticulously would be advisable.

Authors's Reply: Thanks. "r₁" was replaced with "r" for consistency in Equation 2 and Table S2.

Reviewer #1 (Remarks to the Author):

I have read the rebuttal letter and the revised version of the manuscript. The authors have fully addressed my two main concerns.

I consider the manuscript will be of great interest for the readers of Nature Communication.

Reviewer #2 (Remarks to the Author):

I have understood the revisions in the manuscript and the responses, particularly I agree the approach of "data-driven model without using light input" mentioned in the response to my comment (7).

This groundbreaking paper elucidates the balance between the expression patterns of circadian clock genes depended on daylength and the regulation of FT2 expression. This gene expression balance finely controls the level of FT2 expression, and even slight variations of this balance have a dramatic impact on FT2 expression levels (Fig. 3E). This sensitive response has been rigorously reproduced through mathematical modeling. Therefore, the consideration of mathematical models is crucial in understanding the essence of the FT2 expression control mechanism with this "fine balance". However, this "fine balance" may be difficult for readers to understand accurately because it is sensitive and complex.

It appears that GI strongly influences the activation of FT2 expression. In Fig. 3E, it is intriguing that the expression level of GI in Short Day only slightly decreases compared to Long Day (the peak value of Normalized Expression in GI; from 1.0 to 0.9 in Fig. 3E), while FT2 exhibits a significant decrease. This highly sensitive response is of particular interest as it pertains to the control mechanism. Similarly, TOC1 also appears to be highly sensitive to influence (the peak value of Normalized Expression in TOC1; from 0.9 to 1.0 in Fig. 3E). It seems that this narrow range of variation in both GI and TOC1 expression levels harbors the "fine balance"..

In the list of parameters for the mathematical model (Supplementary Table 2), readers can discern the strength of influence of each element. For instance, GI appears to have a strong activation rate ($a = 77.8$), while TOC1 possesses a very strong repression rate ($r = 160.4$).

1) The numerical values for GI activation rate (a), TOC1 Repression rate (r), and LHY2/CDF2 repression activity (K) seem to depend on the "amplitude (maximal expression)" in Equation (1). To facilitate a comparison of the strength of influence of each element, it might be beneficial to normalize the "input(t)" data. Otherwise, LHY2/CDF2 repression activity ($K=4.2$ or 3.1) may appear significantly smaller compared to TOC1's.

2) In the first equation of Equation (2), plotting how the right-hand side first term changes for GI, TOC1, and Repressor concerning the "fine balance" may be achievable. Since the Hill coefficient is 3, it might allow for the clarification of significant nonlinearities in the FT2 gene expression.

Minor Comments:

Are both "a" in Equations (1) and (2) different? If they are distinct, it may be advisable to use different symbols to avoid reader confusion.

Clarify the description of Supplementary Figure 2: Is "LD (red) and SD (blue)" the correct labeling? Please provide the reference/citation for Equation (2).

Reviewer #1

I have read the rebuttal letter and the revised version of the manuscript. The authors have fully addressed my two main concerns. I consider the manuscript will be of great interest for the readers of Nature Communication.

Authors's Reply:

We thank Reviewer for positive evaluation of our work.

Reviewer #2

I have understood the revisions in the manuscript and the responses, particularly I agree the approach of "data-driven model without using light input" mentioned in the response to my comment (7).

This groundbreaking paper elucidates the balance between the expression patterns of circadian clock genes depended on daylength and the regulation of FT2 expression. This gene expression balance finely controls the level of FT2 expression, and even slight variations of this balance have a dramatic impact on FT2 expression levels (Fig. 3E). This sensitive response has been rigorously reproduced through mathematical modeling. Therefore, the consideration of mathematical models is crucial in understanding the essence of the FT2 expression control mechanism with this "fine balance". However, this "fine balance" may be difficult for readers to understand accurately because it is sensitive and complex.

It appears that GI strongly influences the activation of FT2 expression. In Fig. 3E, it is intriguing that the **expression level of GI in Short Day only slightly decreases compared to Long Day** (the peak value of Normalized Expression in GI; from 1.0 to 0.9 in Fig. 3E), while FT2 exhibits a significant decrease. This highly sensitive response is of particular interest as it pertains to the control mechanism. Similarly, **TOC1 also appears to be highly sensitive to influence** (the peak value of Normalized Expression in TOC1; from 0.9 to 1.0 in Fig. 3E). **It seems that this narrow range of variation in both GI and TOC1 expression levels harbors the "fine balance"**.

In the list of parameters for the mathematical model (Supplementary Table 2), readers can discern the strength of influence of each element. For instance, GI appears to have a strong activation rate ($a = 77.8$), while TOC1 possesses a very strong repression rate ($r = 160.4$).

Authors's Reply: We concur that variations in amplitude for GI and TOC1 when comparing SD to LD conditions are subtle (better visualized in Fig. 1E). However, our experimental data revealed a substantial change in the relative phases of these transitioning from LD to SD. In poplar, under SD, GI and LHY2 significantly advance their phase, while TOC1 remains almost unchanged (Fig. 1E). Therefore, in our model, it is not the expression levels but rather the relative gene peak positions that mainly direct the pronounced decrease of *FT2* in SD scenario. In LD, GI phase lags behind TOC1 presumably allowing *FT2* upregulation. In SD, GI and TOC1 phases coincide; as consequence, TOC1 repression, together with the contribution of LHY2 repression,

overcomes GI activation (Fig. 3E; Supplementary Movie 1; Lines 205-210 and 328-340). In Fig. 3E, we applied normalization solely to visualize only the phase shift impact on *FT2* expression dynamics but not in the calculations as stated in the following reply.

1) The numerical values for GI activation rate (a), TOC1 Repression rate (r), and LHY2/CDF2 repression activity (K) seem to depend on the "amplitude (maximal expression)" in Equation (1). To facilitate a comparison of the strength of influence of each element, it might be beneficial to normalize the "input(t)" data. Otherwise, LHY2/CDF2 repression activity ($K=4.2$ or 3.1) may appear significantly smaller compared to TOC1's.

Authors's Reply: Thank you for this comment. We apologize for the confusion. The contributions of LHY2/CDF2 are not lesser than those of GI and TOC1. " K " parameter, now renamed as " $K_{Repressor}$ " (either K_{LHY2} or K_{CDF2} , depending on the model considered), denotes the repression strength, and it is not equivalent to " a/r " activation/repression rates. " a/r " rates are directly related with strength by this formula " $K_x = \sqrt[n_x]{a/r_x} * ka_x$ ". Thus, " $K_{Repressor}$ " combines " $r_{Repressor}$ " and " $ka_{Repressor}$ " into a single value to reduce the number of parameters to fit experimental data. Note that $K_{Repressor}$ is taken to the power $n_{Repressor}$ in equation 2 (see Supplementary Information). This parameter folding cannot be used for TOC1 or GI, as their " ka_x " values also appear in the GI-TOC1 competitive binding term for occupancy of *FT2* promoter - " $(ka_{GI} * ka_{TOC1})^{(n_{GI} * n_{TOC1})}$ ", as well as saturating GI activation for " ka_{GI} ". However, we can directly compare the strengths of the input genes by calculating " K_x " for GI ($K_{GI}= 2.6$) and TOC1 ($K_{TOC1}= 5.4$) as defined above. Therefore, the influence of LHY2 ($K_{LHY2}= 4.2$) and CDF2 ($K_{CDF2}= 3.1$) falls within the same range as GI or TOC1.

These " K_x " strengths may indeed be influenced by gene amplitude without normalization. Nonetheless, our model is based on actual expression levels obtained from qPCR. Omitting this important information about the number of transcripts impacts the simulations of the mutant scenarios, diminishing the biological meaning of the input expression level at each time step (now transcripts per minute) and making more difficult comparisons between experimental and simulated data. Moreover, we believe that normalization would not add much value, and instead have detrimental effect on mutant scenario predictions. In addition, we performed sensitivity analysis to demonstrate how input gene patterns affect *FT2* expression that helps to explore the impact of each clock regulator on *FT2* dynamics (see following answer 2).

2) In the first equation of Equation (2), plotting how the right-hand side first term changes for GI, TOC1, and Repressor concerning the "fine balance" may be achievable. Since

the Hill coefficient is 3, it might allow for the clarification of significant nonlinearities in the *FT2* gene expression.

Authors's Reply: Thanks for your suggestion. In response, we have included a sensitivity analysis in new Supplementary Fig. 10. This analysis evaluates *FT2* expression deviation in area (AUC; transcript accumulation in a day) and amplitude (maximal expression) under LD regime, after altering the expression patterns of the three input genes TOC1, GI, and LHY2. For that, we independently perturbed the three parameters that define the Gaussian input pulse: phase, amplitude, and standard deviation – width (see also equation 1). Briefly, the results confirm that *FT2* expression level is defined by the counteracting actions of GI (promoting) and TOC1 (repressing). We observe maximal *FT2* transcription after losing TOC1 repression before *FT2* peak time, either through TOC1 phase advance or diminished expression (by reducing its amplitude or width), as there is no limit for GI activation until the appearance of LHY2. Similarly, a GI phase delay also increases *FT2*, but to a lesser extent since it is partially restricted by TOC1 repression. Increasing GI amplitude or width promotes unlimited *FT2* expression within the studied range. Furthermore, LHY2 strongly represses *FT2* when their expression times coincide after a phase shift or width increase, preventing *FT2* expression outside its timeframe. However, LHY2 shows a lower effect than TOC1 shaping *FT2* peak under LD, as a reduction in LHY2 amplitude or width results in a more restricted *FT2* promotion. Discussed in the revised manuscript Lines 236-241.

Minor Comments:

Are both "a" in Equations (1) and (2) different? If they are distinct, it may be advisable to use different symbols to avoid reader confusion.

Authors's Reply: Thanks. We have modified the symbols of equations 1 and 2 to prevent misunderstandings.

Clarify the description of Supplementary Figure 2: Is "LD (red) and SD (blue)" the correct labeling?

Authors's Reply: Yes, we have corrected this in the revised version.

Please provide the reference/citation for Equation (2).

Authors's Reply: Equation 2 was elaborated in this work based on well-established ODEs-described GRN models (Karlebach & Shamir, 2008). Clarified in *FT2* pattern simulations section of Supplementary Information.

Karlebach, G., & Shamir, R. (2008). Modelling and analysis of gene regulatory networks. *Nature reviews Molecular cell biology*, 9(10), 770-780.

Reviewer #2 (Remarks to the Author):

I have reviewed the rebuttal letter, the revised manuscript, and the Supplementary Information. The authors have satisfactorily addressed all of my concerns. I believe the manuscript will be of significant interest to the readers of Nature Communications.